# Intestinal Microbiota in Postmenopausal Breast Cancer Patients and Controls

**DOI:** 10.3390/cancers13246200

**Published:** 2021-12-09

**Authors:** Romy Aarnoutse, Lars E. Hillege, Janine Ziemons, Judith De Vos-Geelen, Maaike de Boer, Elvira M. E. R. Aerts, Birgit E. P. J. Vriens, Yvonne van Riet, Jeroen Vincent, Agnes J. van de Wouw, Giang N. Le, Koen Venema, Sander S. Rensen, John Penders, Marjolein L. Smidt

**Affiliations:** 1GROW—School for Oncology and Developmental Biology, Maastricht University Medical Centre, P.O. Box 616, 6200 MD Maastricht, The Netherlands; r.aarnoutse@maastrichtuniversity.nl (R.A.); j.ziemons@maastrichtuniversity.nl (J.Z.); judith.de.vos@mumc.nl (J.D.V.-G.); maaike.de.boer@mumc.nl (M.d.B.); 2Department of Surgery, Maastricht University Medical Centre, P.O. Box 5800, 6202 AZ Maastricht, The Netherlands; emeraerts93@gmail.com; 3Department of Internal Medicine, Division of Medical Oncology, Maastricht University Medical Centre, P.O. Box 5800, 6202 AZ Maastricht, The Netherlands; 4Department of Medical Oncology, Catharina Hospital, P.O. Box 1350, 5602 ZA Eindhoven, The Netherlands; birgit.vriens@catharinaziekenhuis.nl; 5Department of Surgery, Catharina Hospital, P.O. Box 1350, 5602 ZA Eindhoven, The Netherlands; yvonne.v.riet@catharinaziekenhuis.nl; 6Department of Medical Oncology, Elkerliek Hospital, P.O. Box 98, 5700 AB Helmond, The Netherlands; j.vincent@elkerliek.nl; 7Department of Medical Oncology, VieCuri Medical Centre, P.O. Box 1926, 5900 BX Venlo, The Netherlands; yvdwouw@viecuri.nl; 8Department of Medical Microbiology, Maastricht University Medical Centre, P.O. Box 5800, 6202 AZ Maastricht, The Netherlands; giang.le@mumc.nl (G.N.L.); j.penders@maastrichtuniversity.nl (J.P.); 9NUTRIM—School of Nutrition and Translational Research In Metabolism, Maastricht University Medical Centre, P.O. Box 616, 6200 MD Maastricht, The Netherlands; k.venema@maastrichtuniversity.nl (K.V.); s.rensen@maastrichtuniversity.nl (S.S.R.); 10Euregional Microbiome Center, Maastricht University, P. Debyelaan 25, 6229 HX Maastricht, The Netherlands; 11Centre for Healthy Eating & Food Innovation, Maastricht University-Campus Venlo, P.O. Box 8, 5900 AA Venlo, The Netherlands

**Keywords:** gut microbiota, microbiome, oestrogen receptor positive, post menopause, breast neoplasm, faeces

## Abstract

**Simple Summary:**

Besides the already known factors that increase the risk of breast cancer, like hormonal treatment, heredity, and obesity, growing evidence exists that intestinal microbiota can influence breast cancer carcinogenesis. Current clinical information into the role of the intestinal microbiota in breast cancer patients is limited. This study aimed to see whether there are differences in intestinal microbiota richness, diversity, and composition between oestrogen receptor positive breast cancer patients and controls. We concluded that the intestinal microbiota richness, diversity, and composition were not different between breast cancer patients and postmenopausal controls. An increased relative abundance of *Dialister* and Veillonellaceae was observed in breast cancer patients scheduled for adjuvant treatment, which might be caused by a relative decrease in other bacteria due to surgery associated factors rather than an absolute increase. For future studies, we strongly advise a more homogeneous group of breast cancer patients of preferably treatment-naive patients.

**Abstract:**

Background: Previous preclinical and clinical research has investigated the role of intestinal microbiota in carcinogenesis. Growing evidence exists that intestinal microbiota can influence breast cancer carcinogenesis. However, the role of intestinal microbiota in breast cancer needs to be further investigated. This study aimed to identify the microbiota differences between postmenopausal breast cancer patients and controls. Patients and methods: This prospective cohort study compared the intestinal microbiota richness, diversity, and composition in postmenopausal histologically proven ER+/HER2- breast cancer patients and postmenopausal controls. Patients scheduled for (neo)adjuvant adriamycin, cyclophosphamide (AC), and docetaxel (D), or endocrine therapy (tamoxifen) were prospectively enrolled in a multicentre cohort study in the Netherlands. Patients collected a faecal sample and completed a questionnaire before starting systemic cancer treatment. Controls, enrolled from the National Dutch Breast Cancer Screening Programme, also collected a faecal sample and completed a questionnaire. Intestinal microbiota was analysed by amplicon sequencing of the 16S rRNA V4 gene region. Results: In total, 81 postmenopausal ER+/HER2- breast cancer patients and 67 postmenopausal controls were included, resulting in 148 faecal samples. Observed species richness, Shannon index, and overall microbial community structure were not significantly different between breast cancer patients and controls. There was a significant difference in overall microbial community structure between breast cancer patients scheduled for adjuvant treatment, neoadjuvant treatment, and controls at the phylum (*p* = 0.042) and genus levels (*p* = 0.015). *Dialister* (*p* = 0.001) and its corresponding family Veillonellaceae (*p* = 0.001) were higher in patients scheduled for adjuvant treatment, compared to patients scheduled for neoadjuvant treatment. Additional sensitivity analysis to correct for the potential confounding effect of prophylactic antibiotic use, indicated no differences in microbial community structure between patients scheduled for neoadjuvant systemic treatment, adjuvant systemic treatment, and controls at the phylum (*p* = 0.471) and genus levels (*p* = 0.124). Conclusions: Intestinal microbiota richness, diversity, and composition are not different between postmenopausal breast cancer patients and controls. The increased relative abundance of *Dialister* and Veillonellaceae was observed in breast cancer patients scheduled for adjuvant treatment, which might be caused by a relative decrease in other bacteria due to prophylactic antibiotic administration rather than an absolute increase.

## 1. Introduction

Breast cancer is the most common cancer in women worldwide [1]. In the Netherlands, approximately one out of seven women (15%) will develop breast cancer during their lifetime [2]. A combination of genetic, epigenetic, and environmental factors are known to contribute to the development of cancer [3]. Factors such as hormonal treatment, heredity, and obesity are known to increase the risk of breast cancer [4,5]. Still, other factors, such as intestinal microbiota, are thought to influence breast cancer carcinogenesis [4,6,7].

During the last decade, there has been a growing interest in the role of the human intestinal microbiota and the development of cancer. The intestinal microbiota is a collective term for all micro-organisms that colonize the gastrointestinal tract, such as bacteria, yeasts, and fungi. Intestinal microbiota plays an important role in health and disease and carries out crucial functions in the immune system and metabolism of humans [3]. In healthy people, an important equilibrium in the composition of the intestinal microbiota exists, resulting in a personal ecosystem that is essential to maintain homeostasis [8,9]. However, environmental factors and host genetic factors both influence microbiota diversity and composition and can generate dysbiosis leading to a disturbed metabolism and even carcinogenesis [10,11].

One possible link between breast cancer and the intestinal microbiota is the estrobolome, which is defined as the aggregate of intestinal bacterial genes capable of metabolising oestrogens [11]. The enzyme β-glucuronidase has been shown to increase intestinal oestrogen reabsorption into circulation [6]. Specifically, in breast cancer, high levels of circulating oestrogens are related to the development of oestrogen receptor-positive breast cancer [11].

Several pre-clinical and clinical cross-sectional studies have already investigated the dysbiosis of the intestinal microbiota in different types of cancer [12,13,14]. However, few studies have specifically investigated alterations in the intestinal microbiota in breast cancer patients [4,6,7]. A clinical study in breast cancer patients showed that the increased relative abundance of *Blautia* sp. and *Faecalibacterium prausnitzii* was directly related to higher clinical breast cancer stages. In addition, a higher abundance of *Blautia* sp. was associated with higher histoprognostic grades according to according to Scarff-Bloom-Richardson [15]. Only a small number of studies have compared the intestinal microbiota composition in breast cancer patients with a control group. The outcome of these studies has indicated differences in microbiota composition [16,17,18,19,20], of which only one study indicated alterations in the relative abundance of 45 bacterial species between breast cancer patients and controls [19]. However, these studies were limited by their relatively small sample sizes among which the largest included 48 patients and 48 controls [16]. In addition, a cross-sectional study may make it difficult to establish a causal relationship between the two clinical entities. Since there is a direct link between breast cancer risk and high levels of circulating oestrogens, especially in postmenopausal women, it is important to study homogeneous groups concerning pre- and postmenopausal state [21,22].

Current clinical information is limited; therefore, further investigation into the role of the intestinal microbiota in postmenopausal breast cancer patients is required. We hypothesized that the intestinal microbiota richness, diversity, and composition of postmenopausal breast cancer patients differs from postmenopausal women without breast cancer. In this paper, we explored the intestinal microbiota richness, diversity, and composition in postmenopausal women with histologically proven oestrogen receptor positive breast cancer and postmenopausal control women.

## 2. Materials and Methods

### 2.1. Participants

Between November 2017 and February 2020, breast cancer patients were prospectively enrolled in four Dutch hospitals. Eligible patients were postmenopausal women with histologically proven oestrogen receptor positive (ER+) and human epidermal growth factor receptor-2 (HER2) negative breast cancer scheduled for (neo)adjuvant chemotherapy with, adriamycin (doxorubicin), cyclophosphamide, and taxane docetaxel (AC-D), or adjuvant endocrine therapy (tamoxifen). Exclusion criteria included distant metastasis, previous chemotherapy and therapeutic antibiotics use within three months before faecal sampling.

Between September and November 2018, postmenopausal women with negative mammography were enrolled as a control group. Women were screened during the National Dutch Breast Cancer Screening Programme, a national programme that invites women between 50 and 75 years of age every two years for a mammogram. The exclusion criteria for the control group included any type of cancer in history, inflammatory bowel disease, mammography older than 8 weeks, and therapeutic antibiotics use within three months before faecal sampling.

The studies were registered in the Dutch Trial Register (NTR-6296 and NTR-7478) and at ToetsingOnline (NL61646.068.17). All studies were approved by the Medical Ethics Committee azM/UM. The studies were conducted in accordance with the Declaration of Helsinki and Good Clinical Practice. Each participant provided a written informed consent.

### 2.2. Sample and Data Collection

Faecal samples and a questionnaire were collected from all participants. For the breast cancer group, this was done before the start of systemic cancer treatment. After collection, samples were immediately stored in the freezer and transported to the hospital in a cooled transport container (Sarstedt, Nümbrecht, Germany). In the hospital, the samples were stored at −20 °C first and at −80 °C for long-term storage. Questionnaires concerned general medical characteristics among which were weight, length, history of abdominal surgery, smoking, alcohol usage, diabetes, medication use, pro- and prebiotic use, reproductive history, nutritional status, and questions on general performance and wellbeing. Baseline characteristics were registered including Karnofsky performance score (KPS), nutritional status assessed with the Malnutrition Universal Screening Tool (MUST), prophylactic or therapeutic antibiotic administration, prebiotic/probiotic use, exogenous oestrogen use, and the use of nutritional supportive drinks.

### 2.3. Study Endpoints

The primary outcome of this study was to see whether there are differences in microbiota richness, diversity, and composition between breast cancer patients and controls.The secondary outcome for this study was to see whether there are differences in microbiota richness, diversity, and composition between breast cancer patients scheduled for neoadjuvant systemic treatment, breast cancer patients scheduled for adjuvant systemic treatment, and controls.

### 2.4. Faecal Microbiota Analyses

Metagenomic DNA was isolated using the Ambion MagMax^TM^ Total Nucleic Acid Isolation Kit (Thermo Fisher Scientific, Waltham, MA, USA) and consisted of a manual pre-processing procedure followed by automated nucleic acid purification with the KingFisher FLEX (Thermo Fisher Scientific, Waltham, MA, USA). Upon the PCR-amplification of the 16S ribosomal RNA (rRNA) hypervariable V4 gene-region, amplicons were sequenced on a MiSeq platform, as previously described [23].

The bioinformatic analyses of the sequencing data were performed using R studio. For the pre-processing, a standardized in-house pipeline using the software package DADA2 (R version 4.0.3) was applied [24]. Contaminated Amplicon sequence variants (ASVs) identified by decontam [25] were filtered out together with ASVs presented in less than 5% of all samples and a total abundance of less than 0.001%. After filtering, 816 taxa remained in the analysis. For further details on DNA isolation, sequencing, and data pre-processing, see Appendix A.

### 2.5. Statistical Analysis of Clinical Data

Baseline characteristics were analysed in IBM SPSS version 27. For continuous data, normality was tested using the Kolmogorov–Smirnov test. Depending on whether the variable was normally distributed or not, an unpaired *t*-test or the non-parametric Mann–Whitney U test was applied. Levene’s test was used to test for equal variances. For categorical variables, the non-parametric Chi-square test was performed. In the case of low frequencies of binary variables, a Fisher’s exact test was used. Two-tailed tests were used and *p*-values < 0.05 were considered statistically significant. Correlations between differentially abundant taxa and clinical characteristics were measured by means of Kendall’s Tau correlation coefficients (τ).

### 2.6. Statistical Analysis of Intestinal Microbiota Data

Bioinformatic analysis of the sequencing data was performed using R Studio (R version 4.0.0) [26]. Observed species richness and the Shannon index, reflecting microbial diversity, were calculated at the ASV level, using the phyloseq package [27]. Tests of the assumptions of normality and homogeneity of variance as well as subsequent statistical testing were performed as described for clinical data.

The R packages, phyloseq [27], microbiome [28], dplyr [29], ggplot2 [30], and microViz [31] were used for the ordination and visualization of taxonomic composition. Again, taxa present in less than 5% of samples were filtered out for all analyses. Unconstrained ordination was performed using principal component analysis (PCA) based on Aitchison distances at the genus and phylum levels [31]. Permutational multivariate analysis of variance (PERMANOVA) was used to analyse differences in overall microbiota composition between groups with and without adjusting for probiotic use as a potential covariate [31]. The homogeneity of multivariate dispersions was assessed using the microViz package [31] and revealed no significant differences in dispersion. The workflow of ANCOM v2.1, which accounts for the underlying compositional structure and sparseness of microbiota data, was used to identify taxa with differential abundance between controls and breast cancer patients or controls and breast cancer patients treated with adjuvant or neoadjuvant systemic therapy, respectively [32]. We set α < 0.05 at 70% (W) of comparisons as a threshold for significance. Afterwards, bacterial counts were transformed into log^10^(1 + x) abundance by means of the microbiome package [28]. Significant differences identified with the ANCOM v2.1 workflow were confirmed by the Kruskal–Wallis test based on log^10^(1 + x) abundance using IBM SPSS version 27. Subsequently, pairwise comparison with Bonferroni correction for multiple testing was performed. Boxplots were made by means of GraphPad Prism 5 version 5.02.

## 3. Results

In total, 81 postmenopausal ER+ and HER2- breast cancer patients and 67 postmenopausal controls were included. From the breast cancer group, 18 patients were scheduled for neoadjuvant chemotherapy and 63 patients for adjuvant chemotherapy or tamoxifen (Figure 1).

The flow chart presents the number of participants included and the number of faecal samples collected during the study period. Patients scheduled for neoadjuvant systemic treatment were eligible to receive adriamycin (doxorubicin), cyclophosphamide, and docetaxel (AC-D). Patients scheduled for adjuvant systemic treatment were eligible to receive AC-D (*n* = 26) or tamoxifen (*n* = 37). All faecal samples were collected before systemic cancer treatment with chemotherapy or tamoxifen and analysed by the amplicon sequencing of the 16S rRNA V4 gene-region.

### 3.1. Baseline Characteristics

Median age (*p* = 0.929) and BMI (*p* = 0.450) were similar in breast cancer patients and controls. In the year prior to inclusion, 25% of the breast cancer patients compared to 16% of the controls had used therapeutic antibiotics (*p* = 0.236), with a median of 25 weeks since the last antibiotic use for the total group. Within the breast cancer group, 32 patients got prophylactic antibiotics administered perioperative (*p* < 0.001). One patient used prebiotics, and two patients and nine controls used probiotics in the year prior to inclusion (*p* = 0.026). None of the participants used nutritional supportive drinks in the year prior to inclusion. There was no significant difference in the use of past oral contraceptives between breast cancer patients (73%) and controls (72%), with a median use of 15 years in the total group (*p* = 0.288). All other assessed baseline characteristics also did not differ between breast cancer patients and controls (Table 1).

In further detail, the breast cancer group consisted of 18 patients scheduled for neoadjuvant chemotherapy and 63 patients scheduled for adjuvant chemotherapy or tamoxifen after breast cancer surgery. Most tumours consisted of a ductal type (70%), followed by a lobular (21%) and mucinous (7%) type. Patients scheduled for neoadjuvant systemic treatment had a significantly larger clinical tumour size (*p* < 0.001) and a more advanced clinical breast cancer stage (*p* < 0.001). All tumours were oestrogen receptor positive, according to the inclusion criteria. Within the group of patients scheduled for adjuvant treatment, 51% got perioperative prophylactic antibiotics administered (*p* < 0.001) from which the majority of these antibiotics involved cefazolin (Table 2 and Appendix A).

### 3.2. Intestinal Microbiota in Postmenopausal Breast Cancer Patients and Controls

#### 3.2.1. Microbial Richness and Diversity

In total, 148 faecal samples were collected. Faecal samples from breast cancer patients (*n* = 81) were collected before starting neoadjuvant chemotherapy (*n* = 18) or before starting adjuvant chemotherapy or tamoxifen (*n* = 63) (Figure 1). Observed species richness (*p* = 0.561) and the Shannon index (*p* = 0.207) were not different between breast cancer patients and the controls (Figure 2).

#### 3.2.2. Microbial Composition and Community Structure

In the total study population, Firmicutes were the most abundant phylum, followed by Bacteroidota and Actinobacteriota (Figure 3A). At the family level, Lachnospiraceae and Ruminococcaceae were the most abundant (Figure 3B).

While the abundance of microbial genera varied per individual (Figure 3C), unconstrained ordination by means of principal components analysis (PCA) indicated no clustering of samples from the breast cancer group or control group. Similarly, PERMANOVA showed no statistically significant differences in overall microbial community structure at the phylum (*p* = 0.514) and genus levels (*p* = 0.292) between breast cancer patients and controls (Figure 4). Additionally, multivariate analysis using probiotic use as a covariate showed no differences in microbial structure at the phylum (*p* = 0.323) and genus levels (*p* = 0.319) between breast cancer patients and controls. To rule out the possibility that prophylactic antibiotic use in patients scheduled for adjuvant treatment might have masked or confounded potential differences in the microbial community structure between the breast cancer group and control group, we repeated the PERMANOVA analysis upon the exclusion of patients that used prophylactic antibiotics. This additional sensitivity analysis confirmed the lack of differences in microbial community structure between breast cancer patients (*n* = 49) and controls (*n* = 67) at the phylum (*p* = 0.948) and genus levels (*p* = 0.573). In addition, we did not find taxa that were differentially abundant between breast cancer patients and controls at the phylum, family, or genus levels.

### 3.3. Intestinal Microbiota in Breast Cancer Patients Scheduled for Neoadjuvant Systemic Treatment or Adjuvant Systemic Treatment Compared to Controls

Our study population consisted of two groups with a different treatment schedule within the group of breast cancer patients. Patients scheduled for adjuvant systemic therapy (*n* = 63) underwent recent breast cancer surgery before inclusion. Patients scheduled for neo-adjuvant systemic treatment (*n* = 18) did not receive breast cancer surgery yet and had a larger clinical tumour size and a more advanced clinical breast cancer stage as outlined above. Among the patients who were scheduled for adjuvant systemic treatment, 51% (*n* = 32) received intravenous prophylactic antibiotics during the operation according to local protocols for reconstructive breast surgery. To identify the potential influence of these factors on the intestinal microbiota, additional analyses were performed between these groups and the controls.

#### 3.3.1. Microbial Richness and Diversity

Observed species richness (*p* = 0.288) and the Shannon index (*p* = 0.057) were not significantly different between patients scheduled for neoadjuvant systemic treatment, adjuvant systemic treatment, and controls (Figure 5 and Appendix A).

#### 3.3.2. Microbial Composition and Community Structure

PERMANOVA revealed a significant difference in overall microbial community structure between the three groups at the phylum (*p* = 0.042) and genus levels (*p* = 0.015) (Figure 6A). Multivariate analysis, using probiotic use as a covariate showed no influence of probiotic use on the microbial structure at phylum (*p* = 0.404) and genus (*p* = 0.359) levels between patients scheduled for neoadjuvant systemic treatment, adjuvant systemic treatment, and controls. To identify which taxa contributed to the differences in overall microbial community structure, differential abundance analyses were performed. At the phylum level, ANCOM-II analysis did not identify differently abundant taxa between the three groups.

At the family level, Veillonellaceae were found to be significantly different in abundance between the three groups, which was confirmed by a Kruskal–Wallis test (*p* = 0.004) (Figure 6(B_1_); Appendix A). Pairwise comparison with Bonferroni correction for multiple testing identified a higher abundance in patients scheduled for adjuvant systemic treatment compared to patients scheduled for neoadjuvant systemic treatment (*p* = 0.004; Appendix A). No differences in the abundance of Veillonellaceae were found between patients scheduled for adjuvant systemic treatment or neoadjuvant systemic treatment compared to the controls.

Additionally, the abundance of the genus *Dialister* was found to be significantly different between the three groups (Figure 6(B_2_); Appendix A). The significance was confirmed with a Kruskal–Wallis test (*p* = 0.003). Pairwise comparison with Bonferroni correction for multiple testing identified a higher abundance in patients scheduled for adjuvant systemic treatment compared to patients scheduled for neoadjuvant systemic treatment (*p* = 0.003; Appendix A). No differences in the abundance of *Dialister* were found between patients scheduled for adjuvant systemic treatment or neoadjuvant systemic treatment compared to the controls.

To correct for the potential confounding effect of prophylactic antibiotic use in patients scheduled for adjuvant treatment, we repeated the PERMANOVA analysis upon the exclusion of patients that used prophylactic antibiotics. This additional sensitivity analysis indicated no differences in microbial community structure between patients scheduled for neoadjuvant systemic treatment (*n* = 18), adjuvant systemic treatment (*n* = 31), and controls (*n* = 67) at the phylum (*p* = 0.471) and genus levels (*p* = 0.124). This indicates that the previously observed differences disappeared after the exclusion of patients using prophylactic antibiotics. However, a Kruskal–Wallis test revealed that the relative abundance of *Dialister* (*p* = 0.003) and Veillonellaceae (*p* = 0.012) was still significantly different between patients scheduled for neoadjuvant systemic treatment (*n* = 18), adjuvant systemic treatment (*n* = 31) without prophylactic antibiotic use (*n* = 31), and controls (*n* = 67).

#### 3.3.3. Correlations between Differentially Abundant Taxa and Clinical Characteristics

In the whole group, breast cancer stage was negatively correlated to the abundance of Veillonellaceae (*p* = 0.003) and *Dialister* (*p* = 0.007). In line with this, increasing clinical tumour size was associated with a lower abundance of Veillonellaceae (*p* = 0.010). Other clinical characteristics showed no significant correlations with differentially abundant taxa (Table 3).

## 4. Discussion

This prospective cohort study investigated the intestinal microbiota richness, diversity, and composition in postmenopausal women with histologically proven ER+/HER2- breast cancer and postmenopausal controls. Our study showed that microbial richness and diversity in terms of observed species richness and Shannon index and the abundance of specific microbial taxa did not significantly differ between breast cancer patients and controls. The additional analysis of patients scheduled for neoadjuvant systemic treatment, and patients scheduled for adjuvant systemic treatment, and controls also showed no significant differences in microbial richness and diversity. However, at the phylum and genus levels, faecal microbial community structure differentiated the three groups, which was no longer observed after the exclusion of patients scheduled for adjuvant systemic treatment who received prophylactic antibiotics. Significant differences were found in the abundance of *Dialister* and its corresponding family Veillonellaceae between patients scheduled for neoadjuvant systemic treatment and adjuvant systemic treatment.

Regarding the microbiota richness, diversity, and composition in breast cancer patients compared to controls, the availability of clinical studies is limited. Our results showed that there were no significant differences in microbial richness and diversity in terms of the observed species richness and Shannon index between breast cancer patients and controls. These results contrast with two other clinical studies that investigated the association between intestinal microbiota and pre-treatment postmenopausal breast cancer patients [19,20]. Goedert et al. observed a significantly lower microbial richness and diversity, in terms of observed species richness and Chao1 index, in breast cancer patients compared to controls [20]. The opposite results were found in a study by Zhu et al., who showed that breast cancer patients had a higher observed species richness and Chao1 index [19]. In the studies by Goedert et al. and Zhu et al., no distinction was made between different types of breast cancer. This makes the breast cancer group less homogenous than our breast cancer group, which only included ER+/HER2- breast cancer patients.

When examining differences in microbial composition, Zhu et al. found differences the species level [19], whereas Goedert et al. [20] did not find any differences in microbial composition after adjustment for multiple comparisons. Concerning the microbiota composition, our results are more in line with the results of Goedert et al. and suggest that the intestinal microbiota composition is not associated with postmenopausal breast cancer. Even though no differences were found in microbial composition based on 16S rRNA gene sequencing, no conclusion could be drawn concerning the functional potential and activity of the bacteria present. Because the presence of bacteria does not necessarily indicate that they also perform their presumed function. Previous studies have indicated that members of the intestinal microbiota could belong to the so-called estrobolome, which is defined as the aggregate of intestinal bacterial genes capable of metabolising oestrogens [11]. For instance, the bacterial enzyme β-glucuronidase has been shown to increase intestinal oestrogen reabsorption into the circulation [6]. A relationship between intestinal microbiota-related oestrogen metabolism and systemic oestrogen levels has already been demonstrated in small groups [11,33,34]. Specifically, in breast cancer, high levels of circulating oestrogens are related to the development of oestrogen receptor positive breast cancer [11]. To further study microbiota–host interactions, future research should also include functional assessments of intestinal microbiota activity. For example, bacterial β-glucuronidase activity in postmenopausal ER+/HER2- breast cancer patients could be explored by conducting β-glucuronidase activity assays, as described by Biernat et al. [35]. In addition, it will be highly relevant to combine β-glucuronidase activity assay outcomes with high-throughput whole metagenomic shotgun sequencing to determine bacterial metabolic capacity [4].

Significant differences in microbial community structure, as well as an increased abundance of *Dialister* and its corresponding family Veillonellaceae, were observed between patients scheduled for adjuvant systemic treatment compared to patients scheduled for neoadjuvant systemic treatment. Since prophylactic antibiotics were perioperatively administered to 51% of patients scheduled for adjuvant treatment, additional sensitivity analysis was performed including only patients without the use of perioperative prophylactic antibiotics. As a result, no more differences were found in the microbial structure at the phylum and genus levels. This indicates that prophylactic antibiotic use may be responsible for the previously observed differences in microbial community structure between patients scheduled for neoadjuvant systemic treatment, adjuvant systemic treatment, and controls.

In most patients scheduled for adjuvant treatment (43%), cefazolin was administered as a prophylactic antibiotic during surgery. Cefazolin is an antimicrobial active against Gram-positive bacteria and only a few specific Gram-negative bacteria such as *Escherichia coli* and *Proteus mirabilis* [36]. It can therefore be speculated that cefazolin has no direct effect on the *Dialister* and Veillonellaceae, which are Gram-negative stains. This is in line with our results, also indicating differences in the relative abundance of *Dialister* and Veillonellaceae when excluding patients using prophylactic antibiotics. In this context, it needs to be noted that an increased relative abundance of *Dialister* and Veillonellaceae might also be caused by an antibiotic-induced reduction in other bacteria rather than an absolute increase. Therefore, it is recommended that follow-up studies further investigate the effect of prophylactic antibiotics commonly administered during breast cancer surgery and quantify their absolute bacterial abundance by means of qPCR for instance.

However, in addition to prophylactic antibiotic use as a confounder, it is worth considering other factors such as the breast cancer stage, tumour size, and breast cancer surgery, as possible other explanations for the differences between patients scheduled for adjuvant and neoadjuvant therapy in future studies. Breast cancer stage and clinical tumour size were negatively correlated with the abundance of Veillonellaceae and *Dialister* in the whole breast cancer group, which might be caused by the fact that patients scheduled for neoadjuvant systemic treatment had significantly higher clinical breast cancer stages and increased tumour sizes compared to patients scheduled for adjuvant systemic treatment.

To the best of our knowledge, no studies are available that have investigated the effect of breast cancer surgery or extra-gastrointestinal surgery on intestinal microbiota and specifically described the effect of breast cancer surgery on *Dialister* and its corresponding family Veillonellaceae [37]. Similar to the increased Veillonelaceae in our study, increased levels of the pathogenic *Veillonella* of the family Veillonellaceae were observed in colorectal cancer patients after colorectal cancer surgery [38]. Liang et al. (2019) [39] have demonstrated an increased abundance of *Dialister* after gastrectomy in patients with gastric cancer. In addition, the abundances of *Veillonella* and *Dialister* have been shown to be increased after bariatric surgery [40,41]. This may indicate that not specifically breast cancer surgery, but surgery, in general, might modulate the postoperative intestinal microbiota composition. However, the administration of prophylactic antibiotics might also have a confounding effect during other operations. This effect should not be neglected in future microbiota studies as well as in clinical care.

There are several strengths and limitations to this study. Unique to this study is its relatively homogenous study population of postmenopausal ER+ and HER2- breast cancer patients. In addition, sensitivity analyses were performed to reveal the potentially confounding effects of prophylactic antibiotic administration. A limitation of this study is its cross-sectional design, which does not allow the identification of a causal relationship between breast cancer and microbiota changes. Another limitation of the present study can be found in the fact that even if bacteria are present, this does not necessarily mean that they also perform their presumed function. Therefore, we highly recommend the further study of functional microbiota analyses, for instance, β-glucuronidase activity assays and whole metagenomic shotgun sequencing. This will provide more insight into the capacity of the intestinal microbiota to metabolise oestrogen [6,11]. Another limitation is that, although none of the patients received recent systemic cancer treatment, 63 of the 81 breast cancer patients recently received breast cancer surgery. We saw clinical as well as microbial differences between patients who are scheduled for neoadjuvant or adjuvant systemic treatment, which makes the groups more heterogeneous. Ideally, only a group of treatment-naive breast cancer patients who are scheduled for neoadjuvant treatment should be included in future studies to confirm our primary hypothesis. Alternatively, faecal sample collection prior to surgery in patients scheduled for adjuvant systemic treatment is highly recommended. Ideally, future studies should perform longitudinal faecal sampling to assess intestinal microbiota changes over time and relate these to clinical characteristics.

In conclusion, intestinal microbiota richness, diversity and composition were not different between breast cancer patients and postmenopausal controls. The increased relative abundance of *Dialister* and Veillonellaceae was observed in breast cancer patients scheduled for adjuvant treatment, which might be caused by a relative decrease in other bacteria due to prophylactic antibiotic administration rather than an absolute increase. A more homogeneous group of breast cancer patients, consisting of preferably treatment-naive patients, is strongly advised for future studies.

## Figures and Tables

**Figure 1 cancers-13-06200-f001:**
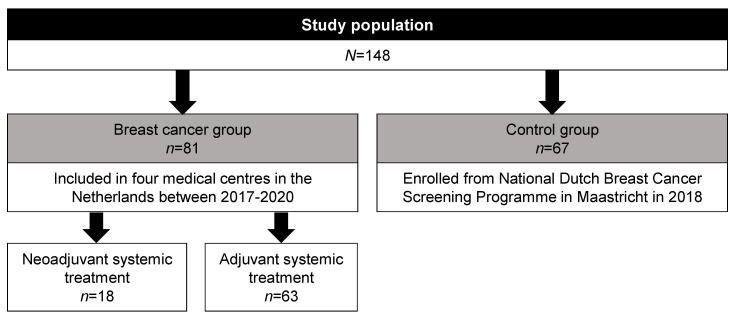
Flowchart study population.

**Figure 2 cancers-13-06200-f002:**
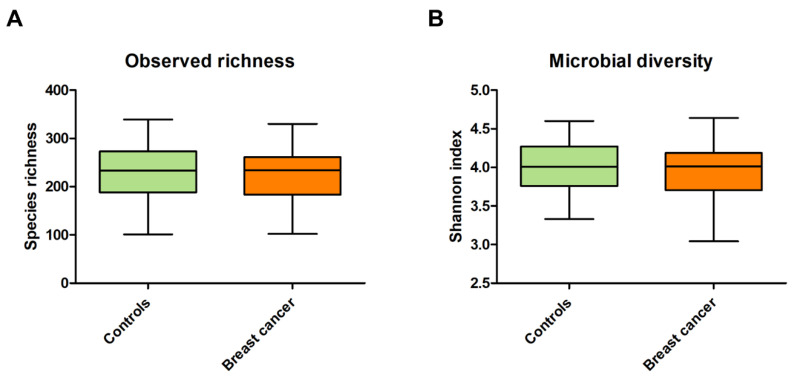
Microbial richness and diversity measures between breast cancer patients and the controls. Observed species richness was analysed with a Mann–Whitney U test (**A**) and Shannon index with an unpaired *t*-test (**B**). For observed species richness, the median and IQR are presented and for the Shannon index the mean and the SD were presented (Appendix A). The observed species richness (*p* = 0.561) and Shannon index (*p* = 0.207) were not different between breast cancer patients and controls.

**Figure 3 cancers-13-06200-f003:**
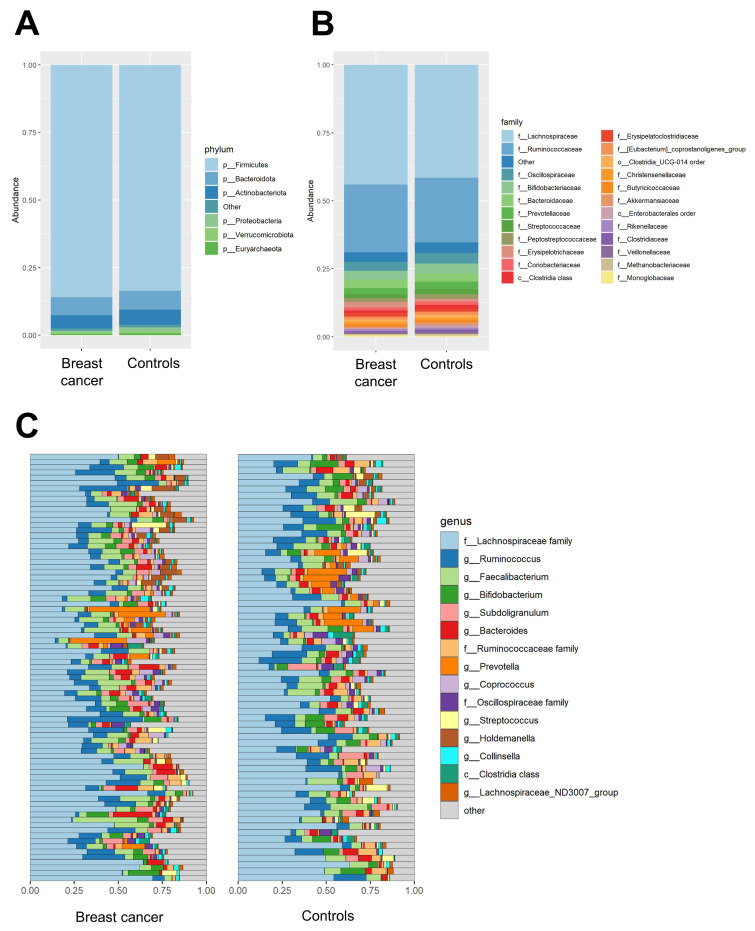
Relative abundances of the most common bacterial taxa in breast cancer patients (*n* = 81) and controls (*n* = 67). (**A**): relative abundance of bacterial phyla with prevalence >10% at the group level; (**B**): relative abundance of bacterial families with a prevalence of >10% at the group level; (**C**): relative abundance of the 15 most common genera in individual patients and controls.

**Figure 4 cancers-13-06200-f004:**
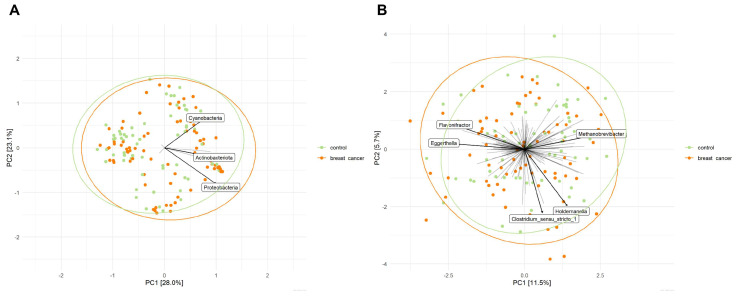
Ordination plots derived from unconstrained principal components analysis (PCA) based on the Aitchison distance, showing the composition of the microbial community at the phylum (**A**) and genus (**B**) levels for breast cancer patients and controls. Taxa present in <5% of the samples were excluded for this analysis. Data were transformed using centre-log-ratio transformation. Names are given for taxa, which contributed most to overall microbial variation.

**Figure 5 cancers-13-06200-f005:**
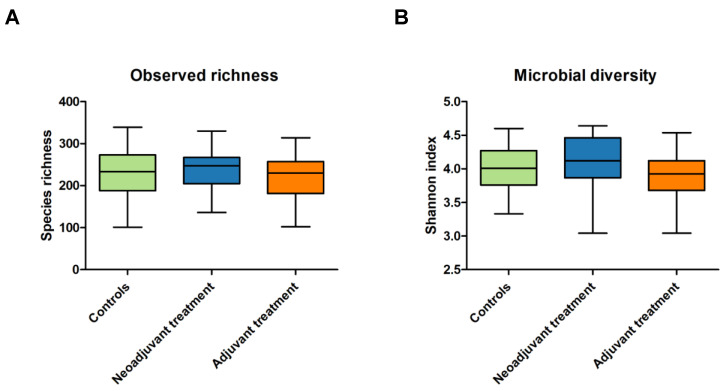
Microbial richness and diversity measures, in terms of observed richness (*p* = 0.288) (**A**) and Shannon index (*p* = 0.057) (**B**), of the patients scheduled for neoadjuvant systemic treatment, adjuvant systemic treatment, and controls analysed with the Kruskal–Wallis test (Appendix A).

**Figure 6 cancers-13-06200-f006:**
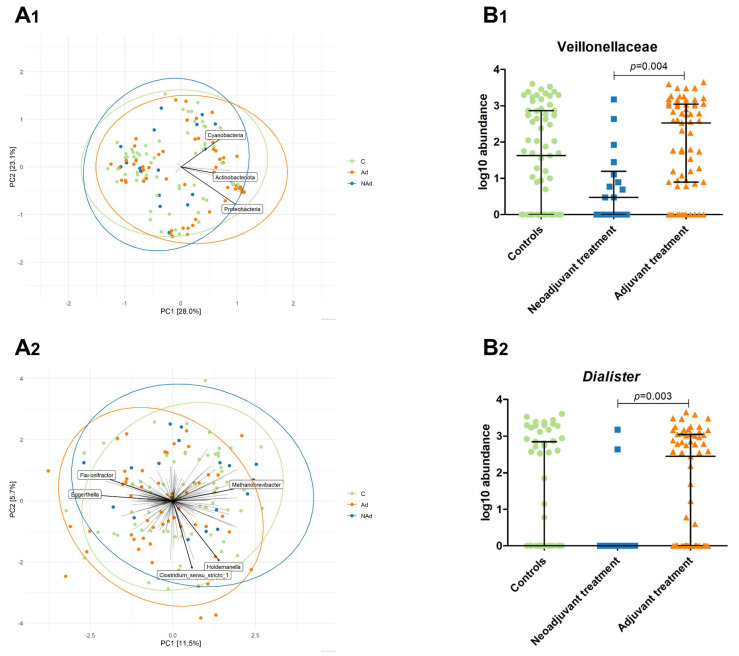
(**A**): Ordination plots derived from unconstrained principal components analysis (PCA) based on the Aitchison distance, showing the composition of the microbial community at the phylum (**A_1_**) and genus (**A_2_**) levels for the neoadjuvant systemic treatment group (*n* = 18), adjuvant systemic treatment group (*n* = 63), and the control group (*n* = 67). Taxa that were present in <5% of the samples were excluded for this analysis. Data were transformed using the centre-log-ratio transformation. Names are given for taxa, which contributed the most to overall microbial variation. (**B**): Scatterplots showing the log^10^ abundance of taxa with significant differential abundance identified with ANCOM-II analyses between patients scheduled for neoadjuvant systemic treatment (*n* = 18), adjuvant systemic treatment (*n* = 63), and controls (*n* = 67). Kruskal–Wallis analyses confirmed significant differences between the three groups identified by ANCOM-II for Veillonellaceae (*p* = 0.004) (**B_1_**) and *Dialister* (*p* = 0.003) (**B_2_**). Adjusted *p*-values in the figures indicated significant differences in log^10^ abundance analysed with pairwise Mann–Whitney U test (Appendix A).

**Table 1 cancers-13-06200-t001:** Clinical characteristics of the study population.

Baseline Characteristics	Total *n* = 148	Breast Cancer *n* = 81	Controls *n* = 67	*p*-Value
Age—years				
*Median (IQR)*	62 (11)	62 (12)	62 (10)	0.929
BMI—kg/m^2^				
*Median (IQR)*	25.3 (5)	25.3 (5)	25.2 (6)	0.450
Karnofsky Performance Score—No. (%) *				
*60*	5 (3)	3 (4)	2 (3)	0.452
*70*	6 (4)	5 (6)	1 (2)	
*80*	29 (20)	17 (21)	12 (18)	
*90*	61 (41)	30 (37)	31 (46)	
*100*	46 (31)	26 (32)	20 (30)	
MUST score—No. (%) *				
*Low risk*	136 (91)	74 (91)	62 (93)	1.00
*Medium risk*	10 (7)	6 (7)	4 (6)	
*High risk*	0 (0)	0 (0)	0 (0)	
*Unknown*	2 (1)	1 (1)	1 (1)	
Diabetes Type 2—No. (%)	11 (7)	7 (9)	4 (6)	0.755
Therapeutic antibiotics use last year—No. (%)	31 (21)	20 (25)	11 (16)	0.236
Duration of antibiotic use last year—days				
*Mean (SD)*	6 (3)	7 (3)	5 (3)	0.154
Time since last antibiotic use—weeks				
*Mean (SD)*	25 (14)	22 (15)	28 (14)	0.337
Prophylactic antibiotic use perioperative—No. (%)	32 (21.6)	32 (39.5)	0 (0)	<0.001
Probiotic use—No. (%)	11 (7.4)	2 (2.5)	9 (13.4)	0.026
Oral contraceptives use past—No (%)	107 (72)	59 (73)	48 (72)	0.889
Oral contraceptives use—years				
*Median(IQR)*	15 (16)	12 (14)	17 (15)	0.288
Time from last oral contraceptives use—years				
*Mean (SD)*	24 (14)	23 (15)	24 (13)	0.582
Time hormonal IUD used—years				
*Mean (SD)*	9 (7)	7 (5)	13 (9)	0.123

* Percentages do not add up to 100% due to rounding. IUD: intrauterine device.

**Table 2 cancers-13-06200-t002:** Clinical characteristics of the breast cancer group.

Clinical Characteristics	Breast Cancer *n* = 81	Neoadjuvant *n* = 18	Adjuvant *n* = 63	*p*-Value
Age—years				
*Mean (SD)*	63 (8)	58 (5)	64 (8)	0.007
BMI—kg/m^2^				
*Median (IQR)*	25.3 (5)	26.2 (7.3)	25.3 (3.5)	0.543
Karnofsky Performance Score—No. (%)				
*60*	3 (3.7)	0 (0)	3 (4.8)	<0.001
*70*	5 (6.2)	0 (0)	5 (7.9)	
*80*	17 (21)	2 (11.1)	15 (23.8)	
*90*	30 (37)	3 (16.7)	27 (42.9)	
*100*	26 (32.1)	13 (72.2)	13 (20.6)	
Breast cancer stage—No. (%) *				
*Stage I*	42 (52)	2 (11)	40 (64)	<0.001
*Stage II*	35 (43)	12 (67)	23 (37)	
*Stage III*	3 (4)	3 (17)	0 (0)	
*Unknown*	1 (1)	1 (6)	0 (0)	
Clinical tumour size (cT)—mm				
*Median (IQR)*	20 (13)	28 (16)	18 (10)	<0.001
Clinical tumour grading—No. (%) *				
*Grade 1*	21 (26)	2 (11)	19 (30)	0.202
*Grade 2*	42 (52)	12 (67)	30 (48)	
*Grade 3*	12 (15)	3 (17)	9 (14)	
*Unknown*	6 (7)	1 (6)	5 (8)	
Tumour focality—No. (%) *				
*Unifocal tumour*	64 (79)	14 (78)	50 (79)	1.000
*Multifocal tumour*	16 (20)	3 (17)	13 (21)	
*Unknown*	1 (1)	1 (6)	0 (0)	
Tumour type—No. (%) *				
*Ductal*	57 (70)	16 (89)	41 (65)	0.051
*Lobular*	17 (21)	2 (11)	15 (24)	
*Mucinous*	6 (7)	0 (0)	6 (10)	
*Unknown*	1 (1)	0 (0)	1 (2)	
PR status—%				
*Median (IQR)*	50 (85)	11.5 (91)	55 (80)	0.218
Time elapsed since operation—days				
*Median (IQR)*	29 (35)	-	29 (35)	-
Prophylactic antibiotic use—No. (%)	0 (0)	0 (0)	32 (50.8))	<0.001
Probiotic use—No. (%)	9 (13.4)	0 (0)	2 (3.2)	0.056

* Percentages do not add up to 100% due to rounding. PR: progesterone receptor.

**Table 3 cancers-13-06200-t003:** Correlations by means of Kendall’s Tau correlation coefficients (τ) between differentially abundant taxa and baseline characteristics.

Baseline Characteristics	*Dialister*		Veillonellaceae	
	Correlation Coefficient	*p*-Value	Correlation Coefficient	*p*-Value
Clinical breast cancer stage	−0.264 **	0.007	−0.272 **	0.003
Clinical tumour grade	−0.072	0.465	−0.104	0.270
Clinical tumour size in mm	−0.156	0.063	−0.204 *	0.010
BMI in kg/m^2^	0.055	0.373	0.050	0.390
Time elapsed since operation in days	0.096	0.295	0.118	0.180
Intravenous prophylactic antibiotic use	0.051	0.606	0.102	0.278

** Correlation is significant at the 0.01 level (2-tailed). * Correlation is significant at the 0.05 level (2-tailed).

## Data Availability

Data is contained within the article or Appendix A. The data presented in this study are available in Appendix A.

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
