# Peer review of "Intestinal Microbiota in Postmenopausal Breast Cancer Patients and Controls"

_cancers, 2021, doi:10.3390/cancers13246200_

Round 1

Reviewer 1 Report

Auhtors shouls adress all comments raised by reviewers or at least provide explanations

Author Response

Dear reviewer,

For a response to the comments made, we would like to refer to the attachment.

Reviewer 2 Report

Authors addressed my questions properly. No comments further.

Author Response

Dear reviewer,

Thank you for reviewing our manuscript.

Reviewer 3 Report

  1. In abstract section page2 line59, why did the authors pick up ER+/HER2-

? Is there any reason for this? We all know there are generally four type of breast cancer, luminal A, luminal B, HER2, and triple negative.

To obtain a homogenous as possible population and since HER2+ breast cancer has an impact on treatment and prognosis, it was decided not to include HER2+ breast cancer patients in this study. Thus, the breast cancer patients in our study involve only luminal A and B. (Above are authors answering my questions.)

  • The authors claimed that HER2+ breast cancer has an impact on treatment and prognosis. So they choose luminal A and B only. However, some luminal B are HER2 positive as well. So I cannot understand the rational here.
  1. For different stage of breast cancer with different duration (Table 2), how could one be sure that these bacteria growth after the breast cancer or the cause of breast cancer?

It is possible to demonstrate a causal relationship between breast cancer and intestinal microbiota changes in a cross-sectional study. In addition, we are aware that the breast cancer group could be more homogeneous based on therapy timing. Currently, the breast cancer group consists of adjuvant and neoadjuvant patients. For future studies, it is an advantage to include neoadjuvant patients over adjuvant patients to eliminate the confounding effect of surgery. We have included these two limitations in our manuscript and clarified that the study does not indicate a causal effect. (Above are authors answering my questions.)

  • Cross-sectional study is well-known to be unable to answer causation questions. So I don’t quite agree with the authors.
  1. In page 3, line76, approximately one out of sever women (15%) will develop breast cancer during their lifetime. This number came from Netherlands. The authors should clarify it. Not all other words area the same. Variations are large. The authors should search and update the most recent worldwide study.
  2. Page 3, line94, “several pre-clinical and clinical cross-sectional studies”, I have checked the reference the authors cites. They didn’t used cross-sectional studies to make sure the causal relationship. This is hard to convince, at least to me.
  3. Also, the control are not matched control criteria. I cannot understand what population the authors refer to. If the initial study groups were from 4 Dutch hospital, the control should be selected from these 4 as well. The description and selection of control are not very clear.

Due to error in study design, I am not going to review further. Thank you very much.

Author Response

(The authors gave the same response as above.)

Reviewer 4 Report

Review cancers-1461514

 Intestinal microbiota in postmenopausal breast cancer patients and controls

Aarnoutse et al. provide a revised manuscript comparing intestinal microbiota on some 148 women nested within prospective trials on breast cancer. I have not had privilege to review the original manuscript. They have not identified differences between controls (breast cancer not detected) vs. those with newly diagnosed breast cancer. Text appears to have numerous revisions judged my highlights.

Study is rich in results, and these are well illustrated. Sufficient reporting of relevant factors influencing microbiome is reported – including probiotic use, and antibiotic use prior to collection.

Other than a rare minor spelling error this study is well conducted and reported.

Author Response

(The authors gave the same response as above.)

Round 2

Reviewer 1 Report

none

This manuscript is a resubmission of an earlier submission. The following is a list of the peer review reports and author responses from that submission.

Round 1

Reviewer 1 Report

The manuscript is well-written and easy to follow. I have some concerns/comments as follows:

  1. There is no power analysis and sample size justification
  2. It is not clear how to identify the controls. Is there a matched algorithm to select the controls?
  3. There is no multivariate model adjusting for confounding variables
  4. A sample size of 18 in Neoadjuvant group is too small
  5. Some variables are missing in Table 2 such as age, BMI and performance score, etc.

Reviewer 2 Report

To the Editor

Thank you for giving me the opportunity to review the manuscript by Prof. dr. M.L. Smidt entitled: “Differences in intestinal microbiota between postmenopausal 1 breast cancer patients and healthy controls”. Authors are to be commended for putting together a concise and well-written manuscript on a very interesting topic.

Major comments:

  1. The population of breast cancer patients included in this study is significantly heterogeneous. Beyond different basal cancer-related characteristics, patients had also significant differences in terms of treatment received and presence of potential confounding factors (i.e. antibiotics administration). Investigating the intestianal microbiota is not an easy task especially taking into account its constant changes and the potential impact of many factors, which even today remain unknown. Thus, in an effort to provide credible results patients enrolled in these studies should be as “homogenous” as possible. This goes also for individuals used as “healthy controls”, at time when such a definition in terms of microbiota is not possible.
  2. Intestinal microbiota analysis in faecal samples is feasible and easy for the patient. However, this methods offers a surrogate only for bacteria in the colon, as fecal specimens draw mainly the distal gut, providing no evidence regarding other sites of the gastrointestinal tract i.e. small bowel that could be implicated (Sarangi A. Methods for Studying Gut Microbiota: A Primer for Physicians. J Clin Exp Hepatol. 2019 Jan-Feb;9(1):62-73).

Below you may find my other comments:

Title

The title of the article should be amended in order to reflect the principal finding of the study. No differences regarding species richness, Shannon index and microbial community structure were evident between the two groups compared; thus, the title should be corrected accordingly to avoid misleading the reader.

Abstract

Page 2, lines 65-66 conclusions: This study was neither designed nor powered to address that “Intestinal microbiota richness, diversity and composition were not associated with postmenopausal breast cancer”. This is rather an assumption and should be omitted.

Introduction

Page 3, lines 74-86: These two paragraphs present the implications of intestinal microbiota alterations in various diseases. However, all these are already known, thus, are redundant information not adding any particular interest. I would suggest to omit them.

Page 3, lines 87-99: Please note that all these studies are cross-sectional in design; thus, any potential causal relationship between the two clinical entities can be hardly established. Instead, all these studies only document alteration in intestinal microbiota composition between breast cancer patients and controls, whenever available. I would suggest to rephrase the Introduction section accordingly.  

Materials and methods – Participants

Page 4, lines 110-122: The cardinal problem when performing instestinal microbiota analysis studies is that this ecosystem is constantly changing and can be affected by many different factors that could eventually alter the final results of the analysis. In this sense, exact strict definition of the populations investigated is of paramount importance. Beyond distant metastasis, previous chemotherapy and therapeutic antibiotics use already listed as exclusion criteria, were there any other exclusion criteria i.e. comorbidities (stroke, chronic obstructive pulmonary disease), systemic diseases (i.e., systemic sclerosis), diabetes mellitus, previous gastrointestinal surgeries with postsurgical structural changes (ileocecal valve resection, gastric bypass, and Roux-en-Y), recent (within the last 3 months) use of proton pump inhibitors (PPI) or H2 blockers, laxatives and drugs that affect intestinal motility (opioids, anticholinergic, and antidiarrheal)? Similarly, please provide a more detailed overview about patients acting as “healthy-controls”. This is perhaps even more conflicting and difficult to define compared to patients but perhaps the most essential part of the study. Thus, I would suggest to provide more clarifications on this issue.

Page 4, lines 128: “questionnaire were collected from all participants”. What kind of questionnaire was this? What was each intention and how it could apply to both cancer patients and healthy ones? Please provide more details.

Page 4, lines 147: Statistical analysis of clinical data. Authors claim in the Introduction section that previous studies were limited due to their small sample size. Did authors made or at least tried to perform a power analysis prior to study initiation?

Please also add a section under the term “study endpoints” where all the primary and secondary outcomes of the study are listed. This way results presentation will follow a specific pattern making the manuscript more “reader-frienfly”.

Results

Page 4, lines 128: How did authors chose the healthy controls from the National Dutch Breast Cancer Screening Programme database? Besides fulfilling inclusion/exclusion criteria, were these individuals i.e. matched for age and gender with the breast cancer patients? This is critical because it raises serious concerns about selection bias, undermining the validity of the results. I would like authors to commend on this. Baseline characteristics i.e. age may not differ but this may be due to chance.

Page 6, lines 187: Baseline characteristics: Please also add information regarding gender

Page 6, Table 1: Please see my comment above regarding diabetes mellitus. This could be a confounder and should be excluded from the study.

Page 6, lines 238: Patients scheduled for neoadjuvant systemic treatment – without the potential confounding effect of surgery on intestinal microbiota – had larger tumour size and more advanced cancer stage; this is certainly an issued highlighting the diversity of the problem and definitions even among patients with similar disease and should be acknowledged as limitation.

Page 12, lines 316-367 Intestinal microbiota in breast cancer patients scheduled for neoadjuvant systemic treatment or adju-316 vant systemic treatment compared to healthy controls

The problem when performing sub-analyses in an already limited sample is that results are susceptible to a type-II error. Although feasible it is not statistically sound and a Bonferroni correction is not a panacea but valuable only when an adequate power analysis for the primary outcome has been conducted. In this sense, authors should at least “tune-down” these sections and data presented here should be cautiously discussed.

Page 12, lines 382-392 Influence of intravenous prophylactic antibiotic administration during operation on the intestinal mi-382 crobiota in breast cancer patients scheduled for adjuvant systemic treatment

This is the most problematic section of the manuscript. Antibiotics administration even at a single dose alter significantly intestinal microbiota composition and thus the results of the study. This is the major limitation of the study and authors should ideally exclude all patients receiving any dose of antibiotics. This would result in a very small sample (n=18) indeed, but this would be also the “real” patient group. Simply by dividing patients scheduled for adjuvant systemic treatment into two groups does not allow exclusion of antibiotic administration as a confounding factor influencing microbiota composition. Instead, it down-sizes the already small and unknown sample constituingn any analysis prone to a type-II error. Thus, I would suggest to totally omit this section.

Page 12, lines 384-385: Authors state that: “prophylactic antibiotics during the operation; this included mostly patients who received breast reconstruction surgery”. Please define “mostly”. How many patients received antibiotics only during the operation and it really possible that antibiotics were not continued to other patients after a major operation? Please also see my comment above.

Discussion

Page 14, lines 487: Authors should also acknowledged as study limitation its cross sectional design that makes difficult to support any causal relationship. In addition, the small sample size and the presence of several confounding factors that could impact their results.

Reviewer 3 Report

The present study described the gut microbiota structure differences between postmenopausal breast cancer patients and healthy people. Overall it is a complete study with well designed procedures. Despite that the Intestinal microbiota richness, diversity and composition were not associated with postmenopausal breast cancer, Dialister and its corresponding family Veillonellaceae were found with differences in cancer patients.

My suggestions are listed below,

The major shortcoming of this study was the Faecal microbiota analyses procedures. I’m worried authors using 16S rRNA V4 gene-region, but not normally used V3, V4 region, which might lead to the incorrect annotating on sequence. In addition, the premiers applied for PCR are missing, which makes me difficult for evaluating on the sequencing quality.

If there are significant differences between the three groups for Veillonellaceae and Dialister, I suggest authors further quantify them based on qPCR to ensure the absolute abundance had same difference.

Reviewer 4 Report

  1. In abstract section page2 line59, why did the authors pick up ER+/HER2-? Is there any reason for this? We all know there are generally four type of breast cancer, luminal A, luminal B, HER2, and triple negative.
  2. Page 3 line 101, “In summary”, it’s kind of strange for in summary appeared in the introduction section.
  3. For different stage of breast cancer with different duration (Table 2), how could one be sure that these bacteria growth after the breast cancer or the cause of breast cancer?
  4. In the introduction section, page 3 line 77, “Still, other factors, such as intestinal microbiota, are thought to influence breast cancer carcinogenesis”. Could the authors write a short paragraph (summary) of what we know currently about microbiota influence(related to ) breast cancer? Is it due to a specific strain?
  5. Many others drugs affects microbiota. Do these studies subject and control all free from other disease, e.g. diabetes, etc. “Crosstalk between gut microbiota and antidiabetic drug action World J Diabetes. 2019 Mar 15; 10(3): 154–168.”
  6. Did their eating habits differ? For example, did these patient like to eat Yogurt? Or did they like to eat meat or vegetarian? Or did they change diet before bacteria collection?

“ Some dietary shifts have the potential to modify the gut microbiota composition and function within the course of days, although the exact time frame might be person-specific, such as the case of dietary fibre supplementation: in some individuals, microbiome alterations were observed as early as 1 day, 2 days, or 3–4 days following supplementation, whereas in others no effects could be noted 3 days, 1 week, 3 weeks or even 12 weeks after such consumption. Likewise, David et al. reported no statistically significant compositional alterations after participants switched to a fibre-rich, plant-based diet for 5 days. By contrast, switching to an animal-based diet rapidly altered the microbiome composition and function, which was reversible upon cessation and might have been attributed to very low intake of fibre or elevated intake of dietary fat and animal protein” from “You are what you eat: diet, health and the gut microbiota. Nature Reviews Gastroenterology & Hepatology volume 16, pages35–56 (2019)”

  1. In page 9, line 316, discuss about the differences of microbiota between patients receiving neoadjuvant, adjuvant and the control groups. It’s hard to imagine that microbiota changes just due to the different treatments approaches. (Page 10, line 346, significantly different between the three groups…)

With beautiful pictures and color, there still exist many confounders. I admire the authors trying to solve this question but couldn’t buying the results.

Reviewer 5 Report

Aarnoutse et al. provide a prospective analysis on intestinal microbiome among some 148 postmenopausal women consisting of both healthy and HR+ Her2 non-amplified breast cancer.

They identified no difference between breast cancer and non-cancer cohorts. They identified subtle difference between those operate and not-operated.

Study is interesting. Conclusion needs a refinement in my opinion.

Comments:

  1. Please describe in METHODS timing of fecal sample collection. It is somehow listed in one Figure caption, but must be better described in METHIDS – for all groups.
  2. Since there is a difference between those operate and not-operated we should be informed how much time elapsed since surgery. Perhaps add to TABLE 1.
  3. Was there any effect on microbiota composition associated with length of time between surgery and sample collection? I.e. “dose-dependent” effect on Veillonellaceae?
  4. RESULTS and Figure 6B suggest that adjuvant therapy group had higher abundance of Dialister. This is also group which was AFTER surgery, but there is  difference in size of tumor. Could be this difference an alternative explanation of Dialister/ Veillonellaceae difference? I do not see that the authors conclusively demonstrated that difference was due to recent surgery – it could be due to a higher tumor burden or other factors driving need for neoadjuvant approach.